



# Declining water resources in response to global warming and changes in atmospheric circulation patterns over southern Mediterranean France

Camille Labrousse[1], Wolfgang Ludwig[1], Sébastien Pinel[1], Mahrez Sadaoui[1], Andrea Toreti[2], Guillaume Lacquement[3]

[1]Centre de Formation et de Recherche sur les Environnements Méditerranéens, Université de Perpignan Via Domitia, CNRS, UMR 5110, 52 Avenue Paul Alduy, 66860 Perpignan Cedex, France
[2]European Commission, Joint Research Centre, Via E.Fermi, 2749, 21027 Ispra, Italy
[3]Acteurs, Ressources, Territoires dans le Développement, Université de Perpignan Via Domitia, UMR 5281, 52 Avenue Paul Alduy, F-66860 Perpignan Cedex, France

*Correspondence to*: Camille Labrousse (Camille.LABROUSSE@ec.europa.eu / camille.m.labrousse@gmail.com)

**Abstract.** Warming trends are responsible for an observed decrease of water discharge in Southern France (northwestern Mediterranean). Ongoing climate change and the likely increase of water demand threaten the availability of water resources over the coming decades. Drought indices like the Reconnaissance Drought Index (RDI) are increasingly used in climate characterization studies, but little is known about the relationships between these indices, water resources and the overall atmospheric circulation patterns. In this study, we investigate the relationships between the RDI drought index, water discharge and four atmospheric teleconnection patterns (TPs) for six coastal river basins in southern France, both for the historical period of the last 60 years and for a worst-case climatic scenario (RCP 8.5) reaching the year 2100. We combine Global and Regional Climate Model (CGM and RCM, respectively) outputs with a set of observed climatic and hydrological data in order to investigate the past relationships between RDI, water discharge and TPs and to project their potential evolutions in space and time. Results indicate that annual water discharge can be reduced by -49/-88% by the end of the century under the extreme climate scenario conditions. Due to unequal links with TPs, the hydro-climatic evolution is unevenly distributed within the study area. Indeed a clustering analysis performed with the RDI time series detects two major climate clusters, separating the eastern and western part of the study region. The former indicates stronger relationships with the Atlantic TPs (e.g. the NAO and the Scand patterns) whereas the latter is more closely related to the Mediterranean TPs (MO and WeMO). The future climate simulations predict an antagonistic evolution in both clusters which are likely driven by decreasing trends of Scand and WeMO. The former provokes a general tendency of lower P in both clusters during spring, summer and autumn, whereas the latter might partly compensate this evolution in the eastern cluster during autumn and winter. However, compared to observations, representation of the Mediterranean TPs WeMO and MO in the considered climate models is less satisfactory compared to the Atlantic TPs NAO and Scand, and further improvement of the model simulations therefore requires better representations of the Mediterranean TPs.



## 1 Introduction

The Mediterranean area was identified as a prominent "hot-spot" for future climate change (Giorgi, 2006). In many areas of
its drainage basin, climate models predict decreasing total precipitation (P) together with increasing temperatures (T) over the
21st century, and in turn a severe threat for surface water resources and river discharges (Q) (Arnell et al., 2011; Pascual et al.,
2015). Water stress may be further exacerbated by growing population (Cramer et al., 2018), and dealing with the increasing
water demand may become a serious challenge.

Atmospheric teleconnection patterns (TPs) explain part of the variability in P, Q and T in the Mediterranean area (e.g. López-
Moreno et al., 2011; Vicente-Serrano et al., 2009; Lopez-Bustins et al., 2008). In the northwestern Mediterranean, climate
conditions are influenced by both large-scale TPs - such as the North Atlantic Oscillation (NAO), the Scandinavian Oscillation
(Scand) and the Mediterranean Oscillation (MO) - and smaller scale TPs, such as the Western Mediterranean Oscillation
(WeMO). Detailed understanding on the relationship between these TPs and the hydroclimatic parameters is therefore key to
assessing future water cycle changes in the Mediterranean area.

Reliable predictions are however complicated by the complex orography which characterises large parts of the Mediterranean
hinterlands. Mountainous areas are important contributors to the annual water discharge for many rivers (Weingartner et al.,
2007), but downscaling of model simulations is difficult in these areas, leading sometimes to opposite trends based on
simulations with different spatial resolutions (Giorgi et al., 2016). Moreover, elevated areas are often exposed to both oceanic
and Mediterranean influence (Molinié et al., 2012), and morphological corridors or barriers control the advection of air masses
from remote humidity sources. The specific evolutions of future surface water resources in response to climate change may
therefore strongly depend on morphology.

In this study, we address these problems by focusing on the past and future evolution of surface water resources in a series of
small coastal river basins in southern Mediterranean France. By looking at the last 60 years, significant negative trends in
annual water discharge series have been identified in several studies (Labrousse et al., 2020; Lespinas et al., 2014, 2010).
Through statistical analysis of the main hydroclimatic parameters and indices, together with the most prominent TPs, we here
demonstrate that the study catchments (despite their rather small extent of 12000 km²) respond differently to the recent
warming trend, due to complex morphological features and exposures to air masses of different origins.

In order to test whether these trends may persist in the future, we further analyse the hydroclimatic evolution of the region till
the end of the 21th century by using simulations from coupled Global and Regional Climate Models (GCMs and RCMs,
respectively) which have been forced by the RCP 8.5 climate scenario. Changes in P and T were converted into changes of
annual water discharge following a set of empirical relationships which have been previously validated (Labrousse et al.,
2020).Furthermore, we demonstrate that the observed decline of surface water resources will continue in all of the study
catchments, although not homogeneously. Strongest reduction will occur in the remote hinterland catchments which are less
impacted by air masses of Mediterranean origins.


## 2 Materials and Methods

### 2.1 Study area

The study area consists of 6 coastal watersheds located in southern France which drain to the Gulf of Lion. From North to South, these are the Herault, Orb, Aude, Agly, Tet and Tech watersheds (Fig. 1). Their area range from 729 km² (Tech) to 4838 km² (Aude). In terms of climatology, they are characterized by a Mediterranean climate type with hot and dry summers, and cool and humid winters. Previous studies demonstrated that these watersheds were already affected by decreasing trends in water discharge over the last decades which could be attributed to recent climatic change (Labrousse et al., 2020; Lespinas et al., 2014, 2010). In terms of morphology, the area is bordered by several mountains ranges which are the Pyrenees in the South and the Haut-Languedoc heights in the North. These mountains play an important role in the climatology of the watersheds. During autumn and winter, Mediterranean cyclonic systems can bring humid air masses from the sea to the hinterlands. Their confrontation with the colder air masses on the mainland is likely to enhance convection process, which can lead to heavy rainfall (Vautard et al., 2015) with hundreds of millimetres of precipitation per day. In terms of land use, the study area is densely populated in the coastal and lowland areas and counts 4 major cities with more than 45 000 inhabitants. The main economic sector is agriculture which strongly relies on the availability of water resources and on appropriate climatic conditions. All these characteristics make that the study area is particularly suitable for examination of the evolution of climatic conditions in the future and their potential impacts on the environment and human activities.

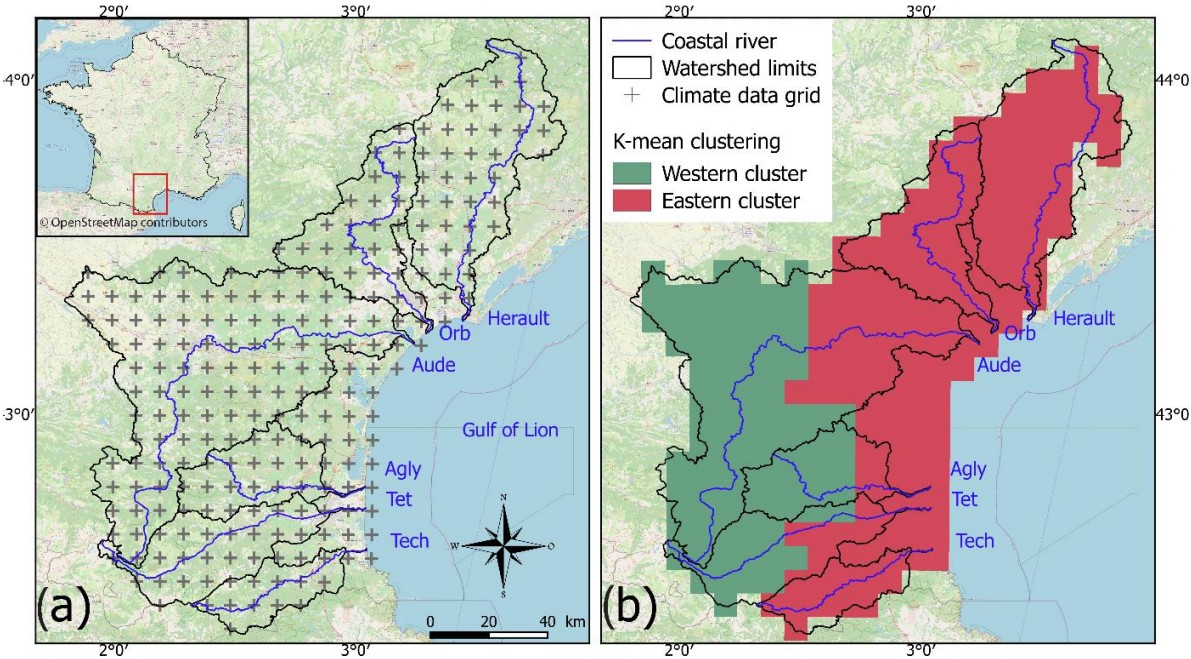

**Figure 1 : Location and characteristics of the study area. a) Limits and rivers of the watersheds and the Safran grid with daily temperature and precipitation b) Cluster distribution obtained by the k-mean method applied to RDI-03 data over the period 1959-2018. Map data: © OpenStreetMap contributors 2021. Distributed under the Open Data Commons Open Database License (ODbL) v1.0.**



## 2.2 Climatic and hydrological parameters

In our study, we use the gridded daily T and P observations provided by Safran on a regular projected grid of 8 km x 8 km for the period 1959-2018 (Fig. 1). Safran is a mesoscale atmospheric system developed by the French meteorological agency Meteo-France (Habets et al., 2008; Soubeyroux et al., 2008). We computed pixel-wise monthly and seasonal averages of each

variable. Watershed and cluster averages (see below) were computed by calculating the mean of all grid points falling within each spatial entity. Boundaries for each watershed were provided by the Carthage database (BD Carthage Métropole, 2021). Although potential evapotranspiration (PET) can be directly extracted from Safran, we reconstructed series of this parameter from temperature data alone according to the formula proposed by Folton and Lavabre (2007) that was validated in our area by previous studies (Labrousse et al., 2020; Lespinas et al., 2014). The reason for this is that PET is not available in simulations

of future climate conditions (see below) and we intended to use a uniform approach for this parameter both for past and future climate conditions. Data of water discharge for each river were taken from the HYDRO database hosted at the French Ministry of Ecology, Sustainable Development and Energy (hydroweb, 2020) which provides daily records for the most downstream gauging stations in the studied river basins. We used the same data series as in Labrousse et al., (2020). Contrarily to our P and T observations, water discharge series do not cover the entire 1959-2018 period, as monitoring of the water gauging

stations only started around 1970, and some of time series are affected by monitoring gaps.

It should finally be mentioned that both for the filling of monitoring gaps and for the prediction of future water discharge series, we applied a statistical multi-parameter hydrological model based on two single climatic indices: RDI-12 and Qpike. RDI corresponds to the drought index of Tsakiris et al., (2007) which is derived from the combination of P and T data. It can be calculated annually (RDI-12) or seasonally (RDI-03). Qpike is based on the combination of annual PET and P data (Pike,

1964) and has been proven to give a realistic estimate of average annual water discharge in Mediterranean rivers (Sadaoui et al., 2018; Ludwig et al., 2009). Labrousse et al. (2020) demonstrated that in all of the six study catchments, multi-regression models fitted with both climatic indices can explain 78-88 % of the variability of annual water discharge during the study period.

## 2.3 K-means clustering

Our study approach is driven by the hypothesis that the climatic and hydrological behaviour of the study area is not uniform, given the differences in morphology and possible connections to air masses of different origins. K-means clustering (Lloyd, 1982; Cam and Neyman, 1967) is a technique that can be used for the detection of climatic regions which behave uniformly and which can consequently be used to test whether different climatic subunits in our study area exist (Fovell and Fovell, 1993; DeGaetano and Shulman, 1990). K-means clustering, as most methods, suffers from an a-priori selection of the number of

clusters and from high dependence on initial conditions. Here we used an elbow heuristic method: the selection of the initial number of clusters is given by a bend in the value of the total within-cluster sum of square (see Bholowalia and Kumar (2014)





). Results and final selection of the number of clusters were done using the function raster.kmeans() from the ecbtools package in R program (Williamson, 2021).

**2.4 Teleconnection patterns**

Monthly historical values for NAO and Scand were taken from the Climate and Prediction Center of the National Oceanic and Atmospheric Administration of USA (NOAA, 2021), while monthly historical values for MO and WeMO were made available by the Climatic Research Unit of the University of East Anglia (MOI data, 2021). For the calculation of future TP projections (see below), we followed the method proposed by Compo et al. (2011) at the Physical Science Laboratory of the NOAA (NOAA Physical Sciences Laboratory, 2021). NAO is consequently defined as the difference in the standardized monthly sea

level pressure anomalies at Lisbon, being the high pressures pole (HP) and Reyjkavik, being the low pressures pole (LP). Similarly and for their positive phase, the Scand pattern has its HP pole over northern Scandinavia (which we defined as the location of the city of Kautokeino, Norway) and has two LP poles located over the southeastern Atlantic (which we defined as the city of Porto, Portugal) and eastern Russia. The MO pattern has its HP over Gibraltar and its LP over Tel Aviv (Israel), and WeMO has a HP pole over San Fernando (Spain) and a LP pole over Padua (Italy). For each standardized series, we

computed the mean sea level pressure surrounded the exact location of each pole (corresponding generally to 4 pixels of each gridded product). Location of the selected poles for each TPs and their computation are given in Table 1.

**Table 1 : Location of the poles chosen for the calculation of each TPs and (b) characteristics of GCMs and RCMs**

| TP | HP location | LP location |
|---|---|---|
| NAO | Lisbon (38.7°N 9°W) | Reykjavik (64°N 22°W) |
| Scand | Kautokeino (69°N 23°E) | Porto (41.2°N 8.6°W) |
| MO | Gibraltar (36.1°N 5.3°W) | Tel Aviv (32.1°N 34.8°E) |
| WeMO | San Fernando (36.5°N 6.3°W) | Padua (45.4°N 11.9°E) |

**2.5 Wavelet analysis**

Wavelet analysis is a powerful method of time series analysis compared with more traditional methods and have been widely used for hydrologic or atmospheric variables since the 1990s (Holman et al., 2011; Liesch and Wunsch, 2019; Holman et al., 2011; Kang and Lin, 2007; Grinsted et al., 2004; Torrence and Compo, 1998). A wavelet is characterized by its localization in both time and frequency and wavelet analysis is therefore an adequate method to examine multiscale phenomena of a climatic series. A review on the detailed applications and objectives of wavelet analysis was made available by Sang (2013). In addition,

the cross-wavelet transform provide a correlation between two signals in the time-frequency space named the common power as well as their relative phase named the continuous wavelet coherence. Cross-wavelet analysis is thus an appropriate method for tracking the relationships between two climatic time series and further work for its application over multivariate climate





series has been recently carried out (Polanco-Martínez et al., 2020). As performed in Liesch and Wunsch (2019), we computed here the wavelet analyses with a Morlet wavelet and based on the monthly data of historical TPs, Q, and RDI-03.

## 2.6 Climate projections

Projected climate data (T, P) under a scenario RCP 8.5 were taken from 6 RCMs which were forced by 4 different GCMs at their boundary conditions. Sea level pressure data were taken from the same 4 GCMs. The data are available at the Copernicus database CMIP5 (Copernicus, 2021) and the characteristics of each GCMs and RCMs are shown in Table 2. RCMs provide daily T and P values which cover the period 1950-2018. By definition, historical simulations span over the period 1959-2005 and can consequently be used to validate the models by comparing the seasonal means of each parameter and their linear trends with the observed data of Safran. Future projections under scenario RCP 8.5 span over the period 2006-2100. Also here, we computed the annual mean for each parameter and performed linear trends to explore their evolution through the projected period. We focus in our study exclusively on the RCP 8.5 which has been released in the fifth assessment report of the IPCC in 2014 (IPCC, 2014). It should be considered as 'worst case scenario' which assumes the greatest fossil fuel use and results in an additional 8.5 watts per square meter of radiative forcing by 2100. Its realism is therefore debated today (see for example Burgess et al., (2020); Hausfather and Peters, (2020); Schwalm et al., (2020)) and the predicted climate evolution should naturally be considered with caution. The main interest of using this "no-climate policy scenario" is that extreme conditions are more suitable for detection of the general trends related to global warming, even if the magnitude of the predicted changes can be exaggerated.

**Table 2 : Characteristics of the GCMs and of the corresponding RCMs**

| GCMs | Institute | Horizontal resolution | Forcing models (Atmosphere, Ocean, Sea ice, Land) | RCMs | Resolution | Institute |
|---|---|---|---|---|---|---|
| IPSL-CM5A-MR | IPSL (France) | 1.25°x1.25° (~138 km) | LMDZ4, ORCA2, LIM2, ORCHIDEE | WRF381P | 0.11°x0.11° (12 km) | IPSL (France) |
| MPI-ESM-LR | Max Planck (Germany) | 1.87°x1.87° (~208 km) | ECHAM6, MPIOM, JSBACH | CCLM4-8-17 | 0.11°x0.11° (12 km) | CLMcom |
| CNRM-CM5 | CNRM and CERFACS (France) | 1.4°x1.4° (~150 km) | ARPEGE-climat, NEMO, GELATO, SURFEX (+TRIP river routing and coupler OASIS 3) | ALADIN63 RACMO22E | 0.11°x0.11° 0.11°x0.11° (12 km) | CNRM (France) KNMI (Netherlands) |
| EC-EARTH | ICHEC (Ireland) | 1.12°x1.12° (~125 km) | IFS, NEMO, LIM2, Htessel, | RCA4 | 0.11°x0.11° (12 km) | SMHI (Sweden) |





## 2.7 Statistics

Single correlation and multiple regression analyses were performed on the basis of the squared Pearson correlation coefficient (Pearson, 1931) which allows quantification of the strength of linear relationships. Linear trend analyses of hydro-climatic
variables were performed using a Mann-Kendall and Sen slope tests (Mann, 1945; Kendall, 1975; Sen, 1968). For the validation of simulated TPs compared to observations, we applied a Tukey's Honest Significance Difference test (Tukey, 1949). Tukey HSD test is a single-step multiple comparison Post Hoc test that is commonly used to assess the significant differences between pairs of group means.

## 3 Results

### 3.1 K-mean clustering

When testing the k-mean clustering algorithm to our study region on the basis of different climatic parameters and different k values (we tested k=2 and k=3), we obtained the most satisfactory results by fixing k to a value of 2 and using the parameter RDI-03, which is based both on T and P data. This results in two clusters which basically separate the study region in an eastern and a western climate cluster (Fig. 1). The eastern cluster is the larger one and includes dominantly the Herault, Orb,
and Tech basins. The western cluster, on the other hand, covers large parts of the upper Aude, Agly, and Tet basins. This feature corroborates with the finding of Labrousse et al. (2020) who reported a generally more elevated warming trend in the former basins during the 1959-2018 period compared to the latter ones, whereas the latter basins depicted a stronger tendency towards decreasing precipitation (although statistically only weakly significant). This indicates that both clusters could behave differently from a climatic point of view. Moreover, it also fits with the findings of Lespinas et al. (2009) who analysed the
seasonality of precipitation in the study region by dividing the 6 river basins in a series of 15 sub-basins. They reported on the basis of the 1965-2004 averages that the strong contrast between high winter and low summer precipitation, as this is required for the Mediterranean climate definition according to Köppen (1936), matches only in the entire Herault and Orb basins as well as in the lower parts of the other basins. The upper parts of the Tech, Tet, Agly and Aude basins have less contrasted seasonality. It is therefore likely that the eastern cluster we identified is more under the influence of local air masses from
Mediterranean origin compared to the western cluster which might be stronger influenced by air masses from remote origin. Notice that especially the Aude basin in the central part of our study region is morphologically connected to the Garonne River basin which drains to the Atlantic. For cluster specific statistical analysis of observed water discharge, we consider in the following that the sum of water discharge of the Herault, Orb and Tech rivers correspond to the eastern cluster whereas the sum of water discharge of the Aude, Agly and Tet rivers correspond to the western cluster.





## 3.2 Wavelet analysis

Univariate wavelet analyses allow an overview of significant periodicities in time series. Figure 2 shows the resulting power spectra for all TPs and selected hydro-climatic variables (RDI-03, Q) within the two clusters. The local maxima of the power spectra are given in Table S1. The represented time series are rather homogeneous and do not depict major break points, which indicates that the hydroclimatic regime did not fundamentally change over the study period. RDI-03 shows generally the strongest yearly cycle with an average power of 15.2 and 16.9 for the eastern and western cluster, respectively. Such high values reflect the fact that this parameter is based on a combination of T and P data and therefore perfectly represents the contrasting seasonal conditions which characterise the Mediterranean climate type. Also water discharge, MO and WeMO depict significant annual cycles but with lower power values. They decrease respectively in the sense in which the parameters are listed. With respect to the clusters, one can notice that power values are generally greater in the western cluster than in the eastern cluster, which means that in the former one there is more regular periodicity than in the latter one.

Also other than annual cycles can be detected. Water discharge in the western cluster further depicts periodicities of 4.3 and 14 years, and in the eastern cluster of 3.0, 8.4, and 9.3 years. Periodicities are hence generally longer in the western cluster than in the eastern one. The large scale TPs NAO and Scand might show both half-year cycles as well as a decade-like cycles ranging from 11 to 16 years (NAO) and from 8 to 10 years (Scand) but those are however not very evident and have been pointed out as such in the study of Chiodo et al. (2019). Finally also for WeMO a long-term cycle of 10 to 20 years (local maximum of 18 years) can be detected which is however, as for NAO and Scand, less significant.

We furthermore performed cross-wavelet analyses between TPs, RDI03 and water discharge for each cluster. The corresponding plots and statistics are presented in the supplementary section (Fig. S1; Table S2). Also here, water discharge of the Western cluster generally show longer cycles of cross wavelet coherence with TPs than in the Eastern cluster. Mediterranean TPs (MO, WeMO) and their coherence with water discharge are more complex compared to Atlantic TPs (NAO, Scand).



**Figure 2 : Continuous wavelet power spectra of the climate indices in the period 1950-2018 for TPs, 1959-2018 for RDI03, and 1976-2018 for Q**

215



### 3.3 Correlation analysis

Correlation analysis between TPs and selected hydro-climatic parameters (RDI-03, water discharge, T, and P) at the seasonal scales is shown in Figure 3. For all parameters, highly significant correlation (or anti-correlation) is detected. The strength and sign of these correlations however strongly depend on the considered season, which again confirm the complex hydroclimatic regime of our analysed region driven by an interplay of air mass fluxes from different origins. For all parameters and all seasons, it can nevertheless be noticed that the eastern cluster has generally higher values of correlation and greater significance levels with TPs than the western cluster. This cluster is obviously more closely connected to the Mediterranean Sea, which is an important reservoir for the atmospheric transfer of water and heat fluxes in our region.

During most of the year, temperature is anti-correlated with Scand and WeMO, which means that during positive phases, colder air masses of northern (Scand) and northwestern (WeMO) origin trigger lower temperatures in the study region. This is valid for both clusters. The influence of Scand is especially dominant during the first part of the year (spring, summer) while the influence of WeMO increases during autumn. Only in winter, the influence of both TPs cease and T is mainly controlled by warm air masses of southwestern origin, as indicated by the positive correlation with NAO and MO.

Humidity fluxes are revealed by the correlation between TPs and P and Q. The patterns associated with both parameters are closely connected, which is also the case for RDI-03 (for which variability in P is obviously more important than variability of T). P is positively correlated in both clusters with Scand throughout the year, except for winter. The colder air masses of northern origin are consequently a source of humidity in the study region. Nonetheless, it should be borne in mind that both the overall formation and maintenance mechanisms of such atmospheric patterns are complex processes (e.g. Wang and Tan, 2020). Only in winter, anti-correlation with NAO becomes dominant for P in the eastern cluster. This anti-correlation of P with NAO is often cited in the literature as one of the major drivers of inter-annual variability of P in the northwestern Mediterranean area (see for example López-Moreno et al., (2011); Vicente-Serrano et al., (2009); López-Moreno and Vicente-Serrano, (2007)). Furthermore, an important modulating influence exists between the Scand and the NAO, which in turn partly affects the climate variability over Europe (Comas-Bru and McDermott, 2014). But here, and for this period analysed, our data indicate that during most of the year the influence of Scand is more important.

Besides correlation with the large scale TPs Scand and NAO, P, Q and RDI-03 are also significantly connected to the Mediterranean TPs MO and WeMO. These relationships are particularly interesting because here, the western and eastern cluster depict large differences, which can at least partly explain why climate in both clusters is different. P in the eastern cluster is significantly anti-correlated with MO in spring, autumn and winter, and with WeMO in autumn and winter. This reflects the arrival of air masses from southern and eastern origins, which could be enriched in humidity when crossing the Mediterranean Sea. Especially the relationship between P and WeMO is worth of being highlighted here. Correlations between P and WeMO switches from positive to negative between the first and the second half of the year. During spring and summer, it is positive (although only significant in summer) for both clusters. This reflects the arrival of humid Atlantic air masses from the north-west. During autumn and winter, however, correlation coefficients switch to highly significant negative values, but





this only holds for the eastern cluster. Negative WeMO index indicates the arrival of Mediterranean air masses from the east,
which can be enriched in humidity when passing over the Mediterranean. As during this period of the year, land surfaces cool
more rapidly than sea surfaces, their arrival is often associated with the formation of violent precipitation and associated flash
floods which are one of the hydrological characteristics in this area (Chazette et al., 2016; Ricard et al., 2012; Nuissier et al.,
2011).

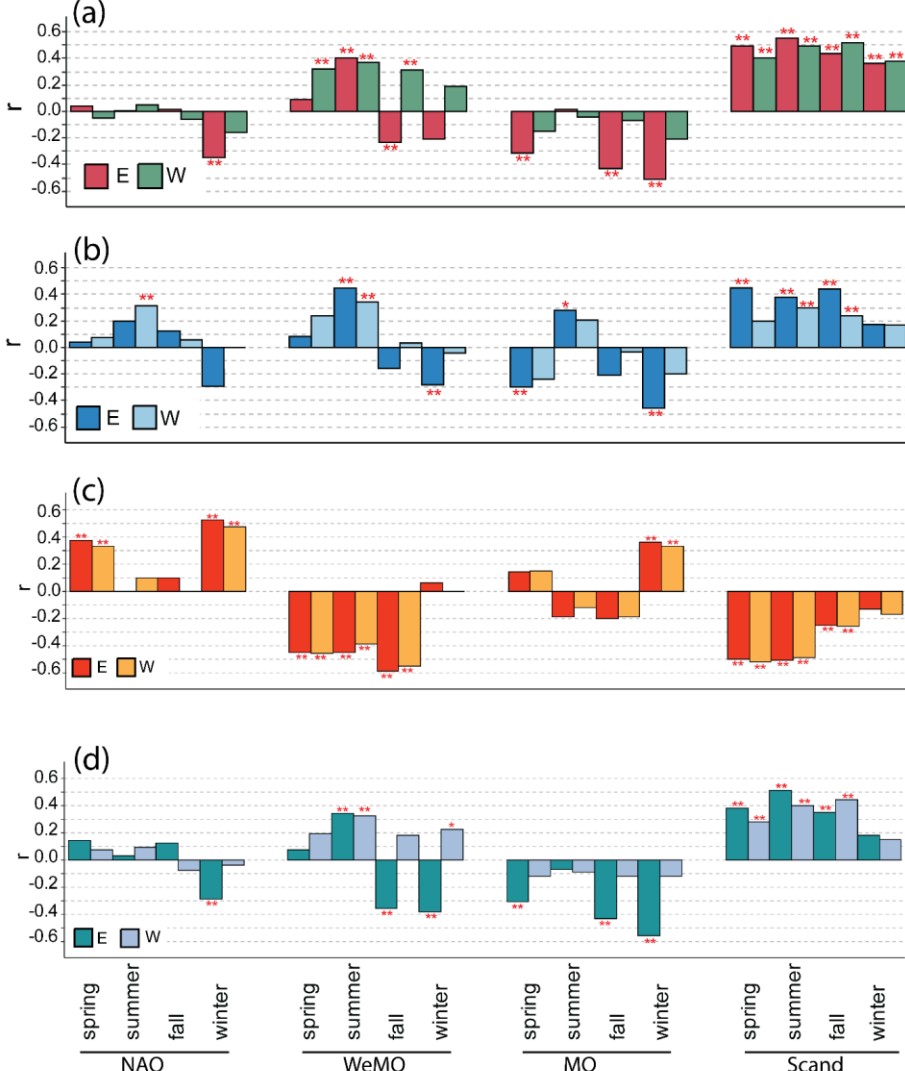

**Figure 3: Estimated correlations between the teleconnection patterns and (a) RDI03 (b) Water discharge (c) Temperature (d)
Precipitation. For (a), (c), and (d) the period considered is 1959-2018, whereas in (b) the period considered is 1976-2018. Water
discharge for each cluster is the sum of each river's water discharge for whose the watershed belongs to either one of the cluster.
Thus, Western discharge is the sum of the Aude, Agly and Tet water discharge and eastern discharge the sum of Herault, Orb and
Tech water discharge**




### 3.4 Variability of TPs in the past and in future climate simulations

During the previous section, we demonstrated that TPs exert a significant control on the hydroclimatic regime of our study region. Before assessing future climate conditions within the RCP 8.5 scenario, we therefore tested first whether there exist a reasonable fit between modelled and observed TPs. Figure 4a illustrates that for the historical period 1950-2005, the four
selected GCMs generally succeed in reproducing the average TP values for NAO, Scand and MO. For NAO, also the observed variability is reproduced in a realistic matter, whereas for Scand, and even more extremely for MO, the simulated variabilities are larger than the observed ones. For WeMO, the opposite holds, which means that the models fail in reproducing the average value and simulate inter-annual variabilities which are lower than the observed ones. With respect to the long-term trends (Fig. 4b), it should be noticed that the models and observations generally follow similar trends for NAO, Scand and MO, but not for
WeMO. Observations for the latter TP depict a strong and significant trend towards lower values during the period 1950-2005 whereas the models produce a rather stationary evolution.

Modelled TPs therefore fit well with observed TPs for NAO, and still reasonably well for Scand. But the fits are less satisfactory for MO and not satisfactory at all for WeMO. Part of these discrepancies may be explained by the fact that WeMO is rather a local scale TP which consequently could suffer from the coarse spatial resolution of GCMs, generally ranging from
100 to 200 km. Notice that the distance between the two poles of WeMO is about 1700 km, compared to 3500 – 4000 km for the other TPs. Another reason may be related to the fact that both WeMO and MO stretch over the (Western) Mediterranean Sea and atmospheric circulation should also be significantly be influenced by the heat and energy content of the Mediterranean water masses. In other words, unless GCMs and RCMs are coupled to detailed oceanic circulation models for the Mediterranean Sea, realistic simulations of WeMO and MO might be difficult.
One may finally also look at the future evolutions of the TPs as they were produced by the GCM simulations (Fig. 4b;Table 3). When averaging the outputs of all models, we find a stationary evolution for NAO and MO, whereas WeMO and Scand follow a significant trend towards lower values. Although the models did not catch the strong decrease of WeMO in the past, they nevertheless predict that the general evolution towards lower values of this TP will persist in the future.

**Table 3 : Linear trend for the annual mean-ensemble teleconnection patterns during the period 2006-2100 and under a scenario**
**RCP 8.5**

| TP | Trend | p-value |
|---|---|---|
| NAO | 0.006 | 0.92 |
| Scand | -0.186 | 0.05 |
| MO | -0.026 | 0.5 |
| WeMO | -0.149 | <0.01 |





**Figure 4: Evolution of the annual teleconnection pattern (a) Variability of teleconnection patterns observed for the period 1950-2005 (Obs) and simulated by the 4 GCMs in the historical period 1950-2005 (Hist). Letters indicate whether box plots are significantly equal (same letter as box plot 'Obs') or not (b) Annual future projections under a scenario RCP 8.5. The blue line represents the annual observed data and is referred as 'Obs' in the legend**



### 3.5 Projections of future climatic conditions

For simulation of the long-term climatic evolution during the period of 1959-2100, we extracted the monthly T and P values
from our RCMs and calculated the multi-model averages. For a control of the ability of our RCMs to reproduce the historical
conditions, we further compare the seasonal averages and linear trends with the observed data extracted from Safran. Figure 5
presents the results of this comparison, as well as corresponding values for the projected period of 2006-2100. The figure
demonstrates that the historical values (1959-2005 averages) almost perfectly fit with the data from Safran for the same period.
This is not really surprising since the RCM data were corrected using the ADAMONT v1.0 method (Verfaillie et al., 2017)
which uses Safran as a forcing function. Both datasets also agree on the seasonal patterns of each cluster, which remain
preserved in the future simulations. For all season, the eastern cluster depicts slightly higher temperatures than the western
cluster, in agreement with its vicinity to the Mediterranean Sea and greater coverage of lowland terrains. For precipitation
however, the relative importance of each cluster is variable according to the season. The western cluster always shows higher
precipitation values during spring and summer compared to the eastern one, whereas the opposite is the case during autumn.
During winter, both clusters have about equal values. Interestingly, in the future simulations, these differences between the
western and eastern clusters slightly decrease during spring and summer, but increase during autumn. This indicates that the
eastern cluster generally becomes more important for catching precipitation in the future.

For trend analyses, the fits between the model simulations and Safran are less satisfactory. During the historical period 1959-
2005, Safran only depicts significant trends for temperature, but not for precipitation. Temperature has strongly increased in
summer (especially in the eastern cluster) followed by spring and by winter. Increase of autumn temperature is the weakest
and statistically not significant. This pattern is in good agreement with the results of Lespinas et al. (2009), who reconstructed
the trends on manual extrapolation of station data for about the same period. However, in the corresponding RCM simulations,
the warming patterns are different. T increased strongest in autumn and in summer, followed by winter and by spring, the
season with the weakest T increase (which is only significant in the eastern cluster). Here, also precipitation follows significant
negative trends, especially in summer and winter in the western cluster.

For the predicted future changes, model simulations predict an important temperature increase, which is strongest in summer
(+5.2 °C) and weakest in spring (+3.9 °C). These increases are almost identical in both clusters. The evolution of precipitation
is characterized by decreasing trends, especially in summer (-37 to -43 %) where the trends are significant for both clusters
and in spring (-5 to -14 %) where the trends are however only significant in the western cluster. Surprisingly, the models
predict even a slight precipitation increase in winter (13 to 17 %) which is however only significant in the western cluster.



**Figure 5: Comparison between Safran and RCMs precipitation and temperature on the historical period 1959-2005 and for the projected period 2006-2100. RCMs-Hist stands for the historical period of RCMs whereas RCMs-Proj is for the projected period 2006-2100 (a)(b) mean annual temperatures and precipitations, respectively (c)(d) Mann-Kendall linear trend of temperatures and precipitations, respectively**





## 4 Discussion

It has been widely established that atmospheric teleconnection patterns like NAO, Scand, Mo and WeMO drive seasonal variability of T, P and Q in the northwestern Mediterranean (see for example Mathbout et al., 2020; Ulbrich et al., 2012; Toreti et al., 2010) and likely will play a significant role in the control their future evolutions (Beranová and Kyselý, 2016). Our data
confirm this for our study region. But there exists a complex interplay between the dominance of each TP which strongly depends on the considered season and the morphological peculiarities. A complete understanding of the hydroclimatic regime in this area is hence highly complex.

However, when clustering the area according to the statistical k-means technique, this understanding can be largely improved. We identified two major clusters which represent respectively the eastern terrains which are more closely connected to air
masses of Mediterranean origins as well as the western terrains which correspond dominantly to the more elevated hinterlands and which are more closely connected to air masses of remote origins. The parameter we used for the cluster separation is the drought index parameter RDI-03. Also other studies demonstrated that drought indexes are suitable for clustering climate sub-units in the Mediterranean area (Manzano et al., 2019). In general, correlations between TPs and hydro-climatic parameters are stronger for the eastern than for the western cluster, especially with regard to the water-flux parameters P and Q. In
mountainous areas, local air convection is important for triggering precipitation (Giorgi et al., 2016; Smith, 1979), which consequently can overprint the influence of large scale air mass exchanges. Also wavelet analysis confirm that both clusters are different. Short and long term periodicity is generally more regular in the western cluster, as revealed by greater power values. Periodicity in the eastern cluster should also be affected by variability of the heat and water mass fluxes of the Mediterranean Sea, which might explain the lower power values in this cluster.
Among the different TPs we tested, Scand and WeMO have obviously the strongest impact on heat and water fluxes in the study region. In its positive phase the Scand is associated with an arrival of air masses from northern Europe which then reach the French Mediterranean coast via a south-easterly flow over the Mediterranean (Kalimeris and Kolios, 2019; Krichak et al., 2014; Bueh and Nakamura, 2007; Lionello et al., 2006). Positive phases of WeMO correspond to a low-pressure centre located over central and eastern Europe and to an anticyclone in the southwest of Spain (Martin-Vide and Lopez-Bustins, 2006), which
favours the arrival of air masses of Atlantic origin in our study region via a north-easterly flow. For precipitation, these remote air masses are generally associated with greater than average values. This holds however only for Scand in the indicated seasons and clusters, whereas in the case on WeMO, the situation is more differentiated. Greater P values in association with positive phases are restricted to the first half of the year (especially summer) but then it switches to a significant negative correlation between both parameters in the eastern cluster. This translates the additional control of this TP on the arrival of
humid air masses of Mediterranean origin, which generate much of P during autumn and winter. WeMO is therefore a significant driver both for humidity sources from Atlantic and Mediterranean origins in our study region.

Interestingly, Scand and WeMO are the two TPs which show significant negative trends in their long-term evolutions of the future climate conditions we constructed on the basis of the extreme RCP 8.5 scenario. The ensemble-mean model simulation





produces a decrease of Scand of about -0.186 between 2006 and 2100 (Table 4). For WeMO this is -0.149. Consequently, for
all of our six study catchments, the future scenarios predict moderate decreases of P (which are however only significant in
the southern Aude, Agly, Tet and Tech catchments), and in combination with the important warming trends, a strong decrease
of Q (Fig. 6). Both P and Q decreases are stronger in the catchments belonging to the western cluster than in those belonging
to the eastern cluster. The latter are also influenced by WeMO during autumn and winter and, as here WeMO is anti-correlated
with P, the long term evolution of WeMO might be associated with relative more precipitation during these seasons, which
can counteract with the general P decrease in relation to Scand.

It should be kept in mind that the values which are depicted in Figure 6 correspond to a worst case scenario and the 55-88 %
reduction of annual water discharge that we found for the individual rivers should naturally be considered with caution.
Moreover, the statistical models we used for the prediction of annual discharge series were calibrated on a narrower range of
temperature increase and it is not clear whether they can be extrapolated to such extreme warming conditions. It should
nevertheless be noticed that also Lespinas et al. (2014) who used in the study region the hydrological model GR2M for a
simulation of water discharge under similar future climate conditions reported that decreases of >80 % could be expected.
There is hence little doubt that future warming should have severe consequences on the availability of surface water resources.
One of the main interest of our future simulation is that it can be directly compared to the evolutions during the recent past in
order to test whether there exist consistent patterns between both evolutions. Both the past (see also Labrousse et al., 2020)
and future evolutions agree in the sense that they depict a general tendency towards lower precipitations in the southern
catchments compared to northern ones, i.e. in the catchments which dominantly constitute the western climate cluster. In the
future scenarios, this is also assigned with consistently greater discharge reductions in these catchments. Of course, the
reliability of these discharge simulations strongly rely on the validity of our statistical models, and, as we only show the
ensemble-mean values, on the variability of simulations between individual models. During the hindcasting period, these
simulations generally fit well with the Safran based simulations (Fig. S2; S3) and all models produce greater discharge
reductions in the western than in the eastern cluster (Table 4). The Safran based hindcasting simulations produce however a
rather equilibrated discharge reduction in both clusters (Labrousse et al., 2020). This is related to that fact that the stronger
precipitation decrease in the western cluster is compensated by a stronger temperature increase in the eastern cluster. Notice
that the GMC-RCM models fail in reproducing these temperature differences and produce a rather homogeneous temperature
increase (Fig. 6). Also seasonally, the models do not well reproduce the warming trends compared to observations (Fig. 5).

The Importance of the WeMO and MO patterns on the western Mediterranean P was first reported by (Gonzalez-Hidalgo et
al., 2009); highlighting that negatives phases of both TPs play in favour for higher P and that predominate the influence of
NAO, which is mainly restricted to the winter season (Dünkeloh and Jacobeit, 2003). We confirm this, but also show that this
mainly holds for the eastern cluster. Unfortunately, compared to observations, the representation of both TPs in the considered
GCM-RCM simulations is less satisfactory than for NAO and Scand, giving less credit to these simulations. This is especially
problematic in the case of WeMO. This TP already showed during the recent past a strong negative trend which is not
reproduced by the considered climate models. In other words, during the forthcoming decades, the decrease of WeMO could





in reality be much more important than the models predict, increasing hence the influence of air masses from Mediterranean origins, and consequently the contrast between both climate clusters. Reliable representation of the Mediterranean TPs in climate models may be more complicated that representation of the Atlantic TPs since this probably requires the coupling with oceanic circulation models at much finer spatial scales in the Mediterranean area in order to catch the heat and water fluxes. For the prediction of the evolution of extreme precipitation events in this area, which is one of major challenges for climate change research, further progress in this field seems to be mandatory.

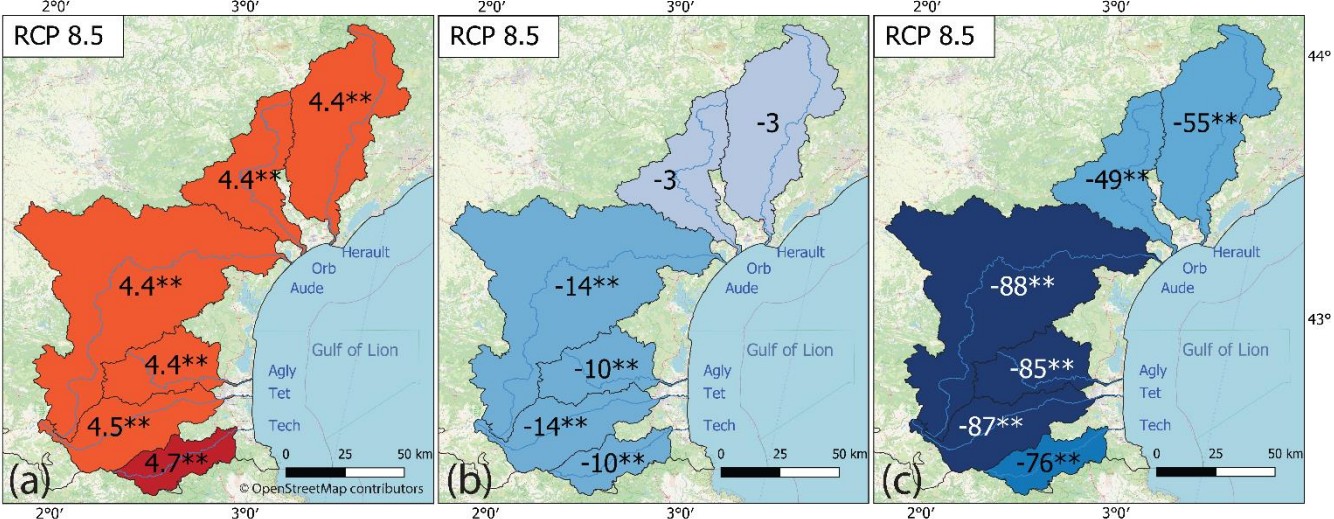

**Figure 6: Linear trends for projected annual hydro-climatic variables under a scenario RCP 8.5 and for the period 2006-2100 (a) temperature in °C (b) precipitation in % and (c) water discharge in %. Map data: © OpenStreetMap contributors 2021. Distributed under the Open Data Commons Open Database License (ODbL) v1.0.**

**Table 4 : Projections of annual series of water discharge by 2100 for each RCMs and according to both clusters**

| RCM | West | East | Difference (%) |
|---|---|---|---|
| ALADIN63 | -87 | -64 | 23 |
| CCLM4-8-17 | -98 | -73 | 25 |
| RACMO22E | -71 | -46 | 25 |
| REMO2009 | -73 | -64 | 9 |
| WRF381P | -100 | -70 | 30 |
| RCA4 | -88 | -63 | 25 |

## 5 Conclusions

The overall goal of this study was to employ an ensemble of CMIP5 model projections to study the annual evolution of water discharge in six coastal Mediterranean rivers in France under a future RCP 8.5 scenario. Relationships between a series of selected hydrological parameters, a drought index and several teleconnection patterns were investigated in order to better
understand the hydroclimatic functioning in the study region during past and future climate conditions. This allows us to come to a number of conclusions which can be summarized as follows:

* Our future simulations on the hydro-climatic evolution in our study catchments confirm that the decrease of surface water resources which has been detected in the recent past is likely to be continued during the forthcoming decades. Under extreme conditions, average annual water discharge could decrease by about -49 % to -88 %. This evolution is mainly driven by the
strong temperature increase which uniformly applies to all catchments, in combination of a moderate precipitation decrease of max. -14 % which is however restricted to the southern catchments.

* There exists a complex interplay between the seasonal evolutions of the major hydro-climatic parameters T, P and Q and the dominant atmospheric TPs NAO, Scand, MO and WeMO. Clustering based on the RDI-03 drought index and the statistical K-means clustering technique however allows the identification of two major climatic clusters which respectively represent
the areas which are under the influence of remote Atlantic (western cluster) and more local Mediterranean (eastern cluster) air masses.

* The future climate simulations predict an antagonistic evolution in both clusters which are significantly related to decreasing trends of the TPs Scand and WeMO. The former provokes a general tendency of lower P in both clusters during spring, summer and autumn, whereas the latter might compensate this evolution by an increase P in the eastern cluster during autumn and
winter.

* Compared to observations, representation of the Mediterranean TPs WeMO and MO in the considered climate models is less satisfactory compared to the Atlantic TPs NAO and Scand. Further improvement of the model simulations therefore requires better representations of the Mediterranean TPs, for example through coupling of high resolution models of oceanic circulation in the Mediterranean Sea.

**Author contribution**

C.L. designed and conducted all experiments and analysed results with advice from W.L., S.P., M.S., and A.T; analyses, C.L., W.L., S.P., M.S.; funding acquisition, W.L., G.L. All authors have read and agreed to the published version of the manuscript.

**Competing interests**

The authors declare that they have no conflict of interest.



**Acknowledgments**

We are especially grateful to Météo-France for supply of the Safran-Isba climate data in the framework of the Publithèque agreement between MF and UPVD.

**Financial support**

This research has been supported by the doctoral school ED 305 at UPVD through the attribution of a PhD grant to CL.

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
