# Peer review of "Declining water resources in response to global warming and changes in atmospheric circulation patterns over southern Mediterranean France"

_Hydrology and Earth System Sciences, 2021_

## Referee Comment (RC2)

**General comments**

The present manuscript proposes to study the link between teleconnection patterns and drought index/discharge over a historical period and under future climate projection. The study focuses on 6 catchments located in southern France.

Although the results presented are interesting and the methods seem adapted, it is not clear what the authors want to put emphasis on. On one hand, the study explains the link between teleconnection patterns, « observed » meteorological variables and « observed » drought index/discharge during the historical period. On the other hand, the authors define a statistical model to produce discharge over future periods, but results and validation only appear during the discussion, which is confusing. The link between the two parts is not clear, making the manuscript sometimes hard to follow.

**Specific comments**

1 Introduction

52-64 : Rather than presenting the results, you could highlight clearly what is done in the article. You could also add a summary of the article.

2 Materials and method

81 : Figure 1, please add the location of the gauging stations.

87-89 : Safran is a reanalysis, a combination of in situ observations and a background (ERA-40 or climatological values for precipitation). See, the definition of the Safran reanalysis in Quintana et al., 2008 and Vidal et al., 2010.

92 : The PET not computed inside the Safran reanalysis. It is computed externally (for example, with the Penman-Monteith formula) or through the combination SAFRAN -ISBA (see Habets et al., 2008)

94 : The variable needed to compute the PET are available in most of the GCM even in future projections. Hence, this is not a sufficient argument to the selection of the formula from Folton and Lavabre (2007).

101-108 : This part could have its own sub-section as this is an important feature of your study.

120 : Please mention the fact that the teleconnection patterns are computed using reanalyses.
https://www.cpc.ncep.noaa.gov/products/wesley/reanalysis.html
https://crudata.uea.ac.uk/cru/data/moi/

122 : Please the fact that the monthly historical values NAO/Scand is not computed with the same formula as the NAO/Scand of the GCM (see the NOAA website).

143 : Please justify the choice of the Morlet wavelet rather than another method.

148 : In Table 2 there are only information about 5 RCM and not 4.

**3 Results**

169 :In order to have a clearer view of the results section could be reorganized (see an example below).

3.1 Analysis of the link between teleconnection patterns and discharge during historical period : K-mean clustering / Wavelet analysis / Correlation analysis / Variability of teleconnection patterns in the past
3.2 Validation of the RCM and empirical hydrological modeling over the historical period
3.3 Analysis under the RCP8.5 scenario : Variability of teleconnection patterns in future climate simulations / Change in T, P and Discharge under future climate simulations / Evolution of discharge between eastern and western cluster

170 : Why did you only try k=2, and k=3? Furthermore, please justify the fact that you used RDI-03 and not another index for the clustering.

217 : In Figure 3, RDI-03 and P seem to give similar results. Can you comment on that.

264-265, Figure 4a : In the text you talk about average values of teleconnection patterns and in the caption of variability. Please clarify this.

281 : It is important to mention the values from different GCM to take into account the uncertainty. Please provide the range of the trends as well as the mean.

288,Figure4a et Figure4b :Teleconnection patterns computed using the reanalyses are noted « Obs », please find another name as teleconnection patterns are not observations nor coming from observations (see comment below).

288,Figure4b : In this figure format, it is hard to really tell if the teleconnections patterns from GCM are coherent with the teleconnections patterns from the reanalyses. It would be better to compare it through a table (as in Table 3) or a boxplot (as in Figure4b). You can include comparison of trend, mean, standard deviation, min, max compute during the 1950-2005 period for both the teleconnection patterns from reanalyses and the teleconnection patterns from GCM.

299 : Please mentioned the correction of RCM using the ADAMONT earlier, in the data presentation (subsection 2.6).

**4 Discussion**

362 : As I mentioned in the general comments, I think it would be better to put this part in the results section and then discuss the results here. Maybe you could still put the result of the projection here, but validation of the hydrological empirical model needs to be addressed earlier.

400, Figure 6 : It is not clear what you are plotting here. In the legend you mentioned linear trends, but it seems rather to be the mean change over the 2006-2100 period in respect to the 1950-2005 period? Furthermore, what is the reference, the simulated discharge the observed discharge.

404, Table 4 :As in Figure 6, it is not clear what you are displaying on the table. Mean changes in annual series?

**5 Conclusion**

Bibliograpy :

Quintana-Segui, P., Moigne, P. L., Durand, Y., Martin, E., Habets, F., Baillon, M., Canellas, C., Franchisteguy, L., and Morel, S.: Analysis of Near-Surface Atmospheric Variables: Validation of the SAFRAN Analysis over France, Journal of Applied Meteorology and Climatology, 47, 92–107, https://doi.org/10.1175/2007jamc1636.1, 2008.

Vidal, J.-P., Martin, E., Franchistéguy, L., Baillon, M., and Soubeyroux, J.-M.: A 50-year high-resolution atmospheric reanalysis over France with the Safran system, International Journal of Climatology, 30, 1627–1644, https://doi.org/10.1002/joc.2003, 2010.

Habets, F., Boone, A., Champeaux, J. L., Etchevers, P., Franchistéguy, L., Leblois, E., Ledoux, E., Moigne, P. L., Martin, E., Morel, S., Noilhan, J., Seguí, P. Q., Rousset   Regimbeau, F., and Viennot, P.: The SAFRAN-ISBA-MODCOU hydrometeorological model applied over France, Journal of Geophysical Research, D06113, 113, https://doi.org/10.1029/2007JD008548, 2008.

---

## Author Comment (AC2)

[Figure]

**Figure 1 Location and characteristics of the study area. a) Limits rivers and gauging stations of the watersheds and the Safran grid with daily temperature and precipitation b) Cluster distribution obtained by the k-mean method applied to RDI-03 data over the period 1959-2018. Map data: © OpenStreetMap contributors 2021. Distributed under the Open Data Commons Open Database License (ODbL) v1.0.**

**Table 1 : Characteristics of the GCMs and of the corresponding RCMs**

| GCMs | Institute | Horizontal resolution | Forcing models (Atmosphere, Ocean, Sea ice, Land) | RCMs | Resolution | Institute |
|---|---|---|---|---|---|---|
| IPSL-CM5A-MR | IPSL (France) | 1.25°x1.25° (~138 km) | LMDZ4, ORCA2, LIM2, ORCHIDEE | WRF381P | 0.11°x0.11° (12 km) | IPSL (France) |
| MPI-ESM-LR | Max Planck (Germany) | 1.87°x1.87° (~208 km) | ECHAM6, MPIOM, JSBACH | CCLM4-8-17 | 0.11°x0.11° (12 km) | CLMcom |
| MPI-ESM-LR | Max Planck (Germany) | 1.87°x1.87° (~208 km) | ECHAM6, MPIOM, JSBACH | REMO2009 | 0.11°x0.11° (12 km) | CSC (Germany) |
| CNRM-CM5 | CNRM and CERFACS (France) | 1.4°x1.4° (~150 km) | ARPEGE-climat, NEMO, GELATO, SURFEX (+TRIP river routing and coupler OASIS 3) | ALADIN63 | 0.11°x0.11° (12 km) | CNRM (France) |
| CNRM-CM5 | CNRM and CERFACS (France) | 1.4°x1.4° (~150 km) | ARPEGE-climat, NEMO, GELATO, SURFEX (+TRIP river routing and coupler OASIS 3) | RACMO22E | 0.11°x0.11° (12 km) | KNMI (Netherlands) |

| EC-EARTH | ICHEC (Ireland) | 1.12°x1.12° (~125 km) | IFS, NEMO, LIM2, Htessel, | RCA4 | 0.11°x0.11° (12 km) | SMHI (Sweden) |
|---|---|---|---|---|---|---|

---

## Author Response (AR1)

**Dear Editor and reviewers**

We appreciate the constructive feedback we received from the two reviewers and have incorporated their suggestions. We have found the comments very useful and feel that they greatly helped improving our manuscript.

We have enclosed a modified version of the manuscript, along with detailed responses to the reviewers' comments.

We hope that you will find this improved version acceptable for publication in the Journal HESS.

Sincerely yours

Camille Labrousse, on behalf of all co-authors

**General comments**

Both reviewers state that the general purpose of our study is not clear enough, making it difficult to follow our scientific argumentation. They further also claim more precision on the statistics we applied and whether our results are supported by physical based knowledge that has been established previously. We have to admit that we can understand these critics since we realize that especially our introduction was not well organized and did not clearly enough emphasize our main study objectives. We therefore largely reworded our introduction in order to change this. It should also help introducing some clarifications about our statistics. In fact, the purpose of our study was not to present the statistical model which was used to translate changes in P and T into changes of Q. This has been done in one of our precious studies (Labrousse et al., 2020). The second part where our statistics were probably not clear enough concerns our k-means analysis, but here we reply more in detail below.

Below are our answers point by point to reviewer 1 (in the following R1) and (R2).

**1 Introduction**

R1 states that the morphological effects in the study area are not well investigated, contrary to what is announced in the introduction. In fact, also here, this may be related to some misunderstanding of what is the purpose of our study. We do not intend a general

investigation about the interactions between climate and morphology. In the case of our study region, morphology is partly responsible for introducing some climatic heterogeneities and we mainly intend to test whether clustering techniques can help to unravel them. We hope that this is much clearer now. Details about the morphological aspects of the study area can be found in section 2.1.

**2 Materials and Methods**

R2 requested to add the gauging stations for each river in Figure 1 in order to add more clarification in the methodology employed.

This was done and can be seen in the revised version of the manuscript in section 2.1

R2 asked on several lines to clarify some terms that were used to describe the kind of data we used.

We answered on those clarifications during the discussion process and incorporated the associated corrections in the revised version of the manuscript. We thank R2 for bringing those details in our work. Please find below the replies we added during the discussion process.

> **Lines 87-89** : This statement will be replaced by « In our study, we use the gridded daily T and P data provided by Safran on a regular projected grid of 8 km x 8 km for the period 1959-2018 (Fig. 1). Safran is a mesoscale atmospheric system developed by the French meteorological agency Meteo-France that uses observation data as well as model outputs for the production of reanalysis data (Durand et al., 1993; Quintana-Seguí et al., 2008). » It can now be found in **Line 92**.

> **Line 92** : We will be more precise on this statement by saying « Although potential evapotranspiration (PET) can be directly extracted from the combination of Safran-Isba, the land surface model which uses the output data from Safran to compute water and surface energy budgets (Soubeyroux et al., 2008; Habets et al., 2008). » This can be found in **Line 97** in the new version of the manuscript.

> **Line 120 :** Here the text will be edited with **«** Reanalysis of monthly historical values for NAO and Scand were taken from the Climate and Prediction Center of the National Oceanic and 120 Atmospheric Administration of USA on the link https://www.cpc.ncep.noaa.gov/data/teledoc/telecontents.shtml. Reanalysis monthly

historical values for MO and WeMO were made available by the Climatic Research Unit of the University of East Anglia on the link https://crudata.uea.ac.uk/cru/data/moi/. » This can be found in the new version of the manuscript at the **Line 140**.

R2 asked why we did not use a PET which was already simulated and available in some models

We answered in the previous step that no PET was made available in the RCM we selected for our study. We used the formula from Folton and Lavabre (2007) because it's the one which was selected and validated for the reconstruction of annual water discharge for historical period in the previous study of Labrousse et al., (2020). This current article is the following of this previous work published in 2020, then it makes it more pertinent to use the same methodology.

R1 requests more details on the definition of the water discharge for each cluster

Water discharge of each cluster is the sum of the water discharge for each river whose delineations of the watershed fall within the limits of a cluster. This is explained in section 2.2, second paragraph. Water discharge data are retrieved from gauging stations located downstream of the rivers and close to the outlet. Data are available in $m^3.s^{-1}$ which we convert per unit area and per year, being thus $mm.year^{-1}$. Water discharge for the western cluster suffer then from an approximation since part of the watersheds belonging to it actually falls in the eastern cluster. Despite this, Figure 3 and the correlation analysis with the teleconnection indices are able to show the same pattern between the water discharge series and the precipitations series with the teleconnection indices.

R1 asks about the initial conditions of the K-means clustering test

The number of clusters and type of the variable analysed were the 2 factors considered to set the initial conditions. The former was based on our *a priori* assumption that at least 2 zones were climatically distinct. This is based on the observations that were made in previous studies by Lespinas et al. (2010, 2014). More details about it have been added in the section 2.3 of the revised version of the manuscript.

R1 about a word in Line 123 of the first version of the manuscript

This was a proofreading mistake, it has been corrected

 requested some clarifications about the definition of univariate wavelet analysis and cross-wavelet analysis

We were pleased to answer this question in the open discussion threat, but we do not think that this precision is necessary for the understanding of the article. On the contrary, we fear that such an explanation, which is rather complex, would hinder the reading of the article. Please find below the answer we provided in the discussion threat.

> "The wavelet transform is a type of mathematical transform that represents a signal according to translated and dilated versions of a finite wave. Compared to a Fourier transform which transforms a time series from its time domain to its frequency domain, the wavelet transform decomposes a signal into a series of wavelets localized both in space and time scales. This type of method provides an efficient approach for the analysis of non-stationary variables such as hydrological and atmospheric time series (e.g., Conte et al., 2021; Sang, 2013; Kang and Lin, 2007). The term « Morlet wavelet » is simply the kind of wavelet we applied. Beforehand, the time-series is compared to a Morlet wavelet transform. The term « cross-wavelet » refers to the analysis of two wavelet transforms from two different time-series together. This one exposes the strength of the common power between the two wavelet spectra as well as the relative phase called coherence of the two wavelet spectra in the time-frequency scale. It can therefore be considered as a correlation coefficient between the two time series."

R2 also asked about the use of a Morlet wavelet

We answered during the discussion process that a Morlet wavelet is a sinusoid modulated by a Gaussian function. It is therefore well suited to detect periodic oscillations at multiple time and frequency scales for real-life signal such as non-stationary climate variables (Labat, 2005; Labat et al., 2005; Torrence and Compo, 1998).

R1 and R2 point out that one model is missing in Table 2

This has been corrected and can be seen in section 2.7 of the revised version.

**3 Results**

In his comments, R2 suggested to rearrange the order the results in order to make the article more clear for the reader

We proposed then this new organization:

3 Results

3.1 Evolution of the climatic conditions and teleconnection patterns over the historical period

      3.1.1 K-mean clustering

      3.1.2 Wavelet analysis

      3.1.3 Correlation analysis

      3.1.4 Variability of TPs

3.2 Validation of the climatic data

3.3 Projection of future hydroclimatic conditions under a scenario RCP 8.5

      3.3.1 Future simulations of TPs

      3.3.2 Future climatic conditions

      3.3.3 Evolution of water discharge

We believe that this new arrangement makes the article more clear in the sense that we start with introducing the different sub-units of distinct climatologies in the study area and their relationships with the TPs in the historical period. Then we show the validation of the modelled data used in this study. And in a third step we present the results of the projections.

R2 asked about the use of the drought index RDI rather than another one

We replied in the previous step that this study is the following of the work presented in Labrousse et al. 2020. The authors used the RDI index to build the statistical model for reconstruction of annual water discharge. For the reason of continuity we maintained the same index in our present study. Otherwise we could not have used our statistical model published previously in order to translate the predicted future changes of P and T into changes of water discharge.

In lines 264-265 R2 suggested to correct the terminology "average values" by "variability"

We agreed with this suggestion and have therefore brought the modification in the new version of the manuscript.

In Table 3, R2 suggested to add the results of linear trend of TPs for each GCMs. It is true that it makes it more transparent and we thus added this modification. It can now be found in section 3.3.1

In Figure 4, R2 suggested to edit the label "Obs" which was used to name the historical data of TPs

We changed this label by "Rea" as it refers to reanalysis data of the TPs

R2 suggested to mention the use of the ADAMONT correction method (used in the correction of the simulated climate data in the output of each RCMs) previously in the manuscript.

We brought this information in the Material and Methods section, in the subsection 2.7

R1 pointed out that the axis in Figure 4 were missing

This Figure shows the evolution of the values for each TPs over the historical and projected period, as well as the variability in a box plot format. For more clarification we therefore added for each y-axis the corresponding TP

R1 asked to clarify the use of the term "more complex" in line 210 in the first version of the manuscript

We answered this statement during the discussion process. Here is our reply

> By « more complex » we mean that the Mediterranean functioning is more irregular than on the Atlantic side. Therefore relationships with water discharge show less long cycles and weaker relationships, as shown also in the correlation analysis. In fact the wavelet analysis here confirms the findings through the K-means since it can show that the eastern cluster has a more irregular behaviour than the western cluster.

**4 Discussion**

In the discussion part, R1 suggested to add sub-sections in order to make it more clear. But after thinking on a way to add it to the manuscript, we found that sub-sections makes the overall discussion less fluent.

R1 indicated that mention of the model REMO 2009, which is shown in Table 4, was missing. This was the missing RCM of Table 2 and it had therefore been added.

R2 asked for clarification of what is shown in Figure 6 as well as in Table 4

Figure 6 shows the linear trends according to a scenario RCP 8.5 over the period 2006-2100. Therefore, the reference period is the beginning of the time series. For water discharge, only simulated water discharge is computed here, from 2006 to 2100. Coherence of the statistical

model used for the computation of the simulated series has been tested and validated in the study of Labrousse et al. 2020. Moreover, additional explanations have been added with regard to Figure 6 in **Lines 355-359** of the revised version of the manuscript. Table 4 shows the linear trends for each RCMs and for each cluster. It means that we computed the trends (in %) for each RCMs and show the average results for the watersheds belonging to each cluster respectively.

R2 also suggested to shift the results of the projections of the climatic data to the discussion and to debate the validation of the empirical hydrological model earlier, in the results section. But the experimentation and use of the empirical hydrological model is one of the central points of the study of Labrousse et al. 2020. And this model was implemented on the same six rivers that we investigate in this current study. Thus, for more details on the validation of the model, we refer to the previous of Labrousse et al. 2020 since we cannot show again in this study the results obtained in the previous one. However, we added in the Results section two sub-sections (3.3.2 and 3.3.3) in which we show separately the results of the projections for the climatic conditions and for the hydrological conditions. We also specified in sub-section 3.3.3 that we used the methodology implemented in Labrousse et al. 2020.

**5 Conclusion**

The conclusion has been reworked in the revised version of the manuscript in order to make it clearer with regard to the objectives described in the introduction.

**References**

Conte, M., Contini, D., and Held, A.: Multiresolution decomposition and wavelet analysis of urban aerosol fluxes in Italy and Austria, Atmospheric Research, 248, 105267, https://doi.org/10.1016/j.atmosres.2020.105267, 2021.

Durand, Y., Brun, E., Merindol, L., Guyomarc'h, G., Lesaffre, B., and Martin, E.: A meteorological estimation of relevant parameters for snow models, 18, 65–71, https://doi.org/10.3189/S0260305500011277, 1993.

Habets, F., Boone, A., Champeaux, J. L., Etchevers, P., Franchistéguy, L., Leblois, E., Ledoux, E., Moigne, P. L., Martin, E., Morel, S., Noilhan, J., Seguí, P. Q., Rousset-Regimbeau, F., and Viennot, P.: The SAFRAN-ISBA-MODCOU hydrometeorological model applied over France, 113, https://doi.org/10.1029/2007JD008548, 2008.

Kang, S. and Lin, H.: Wavelet analysis of hydrological and water quality signals in an agricultural watershed, Journal of Hydrology, 338, 1–14, https://doi.org/10.1016/j.jhydrol.2007.01.047, 2007.

Labat, D.: Recent advances in wavelet analyses: Part 1. A review of concepts, Journal of Hydrology, 314, 275–288, https://doi.org/10.1016/j.jhydrol.2005.04.003, 2005.

Labat, D., Ronchail, J., and Guyot, J. L.: Recent advances in wavelet analyses: Part 2—Amazon, Parana, Orinoco and Congo discharges time scale variability, Journal of Hydrology, 314, 289–311, https://doi.org/10.1016/j.jhydrol.2005.04.004, 2005.

Labrousse, C., Ludwig, W., Pinel, S., Sadaoui, M., and Lacquement, G.: Unravelling Climate and Anthropogenic Forcings on the Evolution of Surface Water Resources in Southern France, 12, 3581, https://doi.org/10.3390/w12123581, 2020.

Lespinas, F., Ludwig, W., and Heussner, S.: Impact of recent climate change on the hydrology of coastal Mediterranean rivers in Southern France, Climatic Change, 99, 425–456, https://doi.org/10.1007/s10584-009-9668-1, 2010.

Lespinas, F., Ludwig, W., and Heussner, S.: Hydrological and climatic uncertainties associated with modeling the impact of climate change on water resources of small Mediterranean coastal rivers, Journal of Hydrology, 511, 403–422, https://doi.org/10.1016/j.jhydrol.2014.01.033, 2014.

Quintana-Seguí, P., Le Moigne, P., Durand, Y., Martin, E., Habets, F., Baillon, M., Canellas, C., Franchisteguy, L., and Morel, S.: Analysis of Near-Surface Atmospheric Variables: Validation of the SAFRAN Analysis over France, J. Appl. Meteor. Climatol., 47, 92–107, https://doi.org/10.1175/2007JAMC1636.1, 2008.

Sang, Y.-F.: A review on the applications of wavelet transform in hydrology time series analysis, Atmospheric Research, 122, 8–15, https://doi.org/10.1016/j.atmosres.2012.11.003, 2013.

Soubeyroux, J.-M., Martin, E., Franchisteguy, L., Habets, F., Noilhan, J., Baillon, M., Regimbeau, F., Vidal, J.-P., Moigne, P. L., and Morel, S.: Safran-Isba-Modcou (SIM) : Un outil pour le suivi hydrométéorologique opérationnel et les études, PP. 40-45, 2008.

Torrence, C. and Compo, G. P.: A Practical Guide to Wavelet Analysis, 79, 61–78, https://doi.org/10.1175/1520-0477(1998)079<0061:APGTWA>2.0.CO;2, 1998.

---

## Referee Report (RR1)

The present manuscript proposes to study the link between teleconnection patterns and discharge over a historical period and their evolution in future climate projection. The study focuses on six catchments located in southern France divided into two sub-regions using clustering method.

The first round of revision has made the goal of the paper much clearer and allows to clearly distinct validation and application.

One focus of the study is the distinction between two sub-regions based on clustering method. The distinction between those two regions is well discussed and seem of interest. However, most of the time the results are presented in the global region: correlation between Q, T, P and the TP (Section 3.1.3 Correlation analysis), evolution of the TP during the historical and future periods (Section 3.1.4 Variability of TPs in reanalysis and GCM simulations and Section 3.3.1 Future evolutions of TPs).

433: GMC-RCM should be GCM-RCM

---

## Author Response (AR2)

**Dear Editor and reviewers**

We are grateful for the valuable comments of our referees (in the following R1 and R2) about our manuscript « Declining water resources in response to global warming and changes in atmospheric circulation patterns over southern Mediterranean France » that my co-authors and I submitted for publication to HESS. The comments were very useful and helped improving our manuscript.

We have enclosed a revised version of the manuscript, along with a detail response of the latest comments.

We hope that you will find this improved version acceptable for publication in the Journal HESS.

Thank you

Camille Labrousse, on behalf of all co-authors

**General comments**

Both R1 and R2 acknowledged that the quality of the manuscript improved after a first revision including a general reorganization of its sections. Both referees agreed on the scientific significance of our study (R2 gave the maximum rate of Excellent) and recognized a good presentation quality. As such not much comments were presented by the referees. R1 proposed a straight publication of the manuscript. R2 however suggested a clarified presentation of the results for each cluster that we (the authors) define in section 2.4 (K-means clustering). We acknowledge that such a presentation is missing in section 3.3.2 (which shows results for the future climatic projections) but believe that it is also important to show the evolution of the hydro-climatic conditions at the watershed scale. As described in section 2.1 (Study Area), the morphological features of each watershed play a non-negligible role in their climatic and hydrological functioning. For this reason, and in order to avoid important reorganisation of the corresponding text parts, we propose to keep the Figure 6 as it was presented in the previous version of the manuscript which shows the future evolution through linear trends of the hydro-climatic conditions for each watershed. But we produced an extra figure in the supplementary materials section (Figure S4) which further summarizes the hydro-climatic evolution at the cluster scale. Presentation and discussion of these results are given in the main text sections

3.3.2 and 3.3.3 (lines 358-361 and line 375), and in the legend of the supplementary Figure S4. This new figure is also provided in a separate pdf file.